# Micro Nanofibrillated Cellulose as Functional Additive Supporting Processability of Surface-Active Mineral Suspensions: Exemplified by Pixel Coating of an NO*x*-Sorbent Layer

**DOI:** 10.3390/ma16041598

**Published:** 2023-02-14

**Authors:** Katarina Dimic-Misic, Monireh Imani, Nemanja Barac, Djordje Janackovic, Petar Uskokovic, Ernest Barcelo, Patrick Gane

**Affiliations:** 1Department of Bioproducts and Biosystems, School of Chemical Engineering, Aalto University, 00076 Helsinki, Finland; 2Innovation Centre of Faculty of Technology and Metallurgy Ltd., Karnegijeva 4, 11200 Belgrade, Serbia; 3Faculty of Technology and Metallurgy, University of Belgrade, Karnegijeva 4, 11200 Belgrade, Serbia; 4Omya International AG, Baslerstrasse 42, 4665 Oftringen, Switzerland

**Keywords:** rheology of undispersed mineral suspensions, pixelated coating, surface flow filtration, micro nanofibrillated cellulose (MNFC)

## Abstract

Unlike established coating formulations, functional particulate coatings often demand the omission of polymer dispersant so as to retain surface functionality. This results in heterogeneous complex rheology. We take an example from a novel development for an NO*_x_* mitigation surface flow filter system, in which ground calcium carbonate (GCC), applied in a coating, reacts with NO_2_ releasing CO_2_. Inclusion of mesoporous ancillary mineral acts to capture the CO_2_. The coating is applied as droplets to maximize gas-contact dynamic by forming a pixelated 2D array using a coating device consisting of protruding pins, which are loaded by submersion in the aqueous coating color such that the adhering droplets are transferred onto the substrate. The flow is driven by surface meniscus wetting causing lateral spread and bulk pore permeation. Filamentation occurs during the retraction of the pins. Stress-related viscoelastic and induced dilatancy in the suspension containing the ancillary mesoporous mineral disrupts processability. Adopting shear, oscillation and extensional rheometric methods, we show that the inclusion of an ancillary mineral that alone absorbs water, e.g., perlite (a naturally occurring porous volcanic glass), is rheologically preferable to one that in addition to absorbing water also immobilizes it on the mineral surface, e.g., sepiolite. When including micro-nanofibrillated cellulose (MNFC), critical for maintaining moisture to support NO_2_ sorption, it is observed that it acts also as a flow modifier, enabling uniform coating transfer to be achieved, thus eliminating any possible detrimental effect on mineral surface activity by avoiding the use of soluble polymeric dispersant.

## 1. Introduction

The requirement for advanced functional coatings leads increasingly to the adoption of component particles that, themselves, are bulk chemically or surface reactive, and this reactivity is designed to impart the necessary functionality of the coating. Colloidal suspensions employed for such coatings frequently contain particulate materials, which are mineral-based. In cases where the mineral surface itself is not the main characteristic required for the coating design, their suspension in water is typically formulated to maintain homogeneous particle-particle electric charge repulsive dispersion, according to the static potential field theory proposed by Derjaguin, Landau, Verwey, and Overbeek, commonly known by combining the first letters of their names, i.e., the DLVO model [1,2]. This is often achieved using anionic, or less often cation or steric, dispersant(s) adsorbed onto the particle surface. A frequently used anionic dispersant is sodium polyacrylate. The polymer chain adsorbs either by balancing existing oppositely charged sites on the particle surface or by undergoing cation exchange followed by coagulation onto the particle surface. For example, exchanging Na^+^ via the chelating action of polyacrylate toward Ca^2+^ enables the dispersion of calcium carbonate particles (CaCO_3_) following the coagulation of the chelate onto the particle surface [3]. The resulting uniform anionic charge then acts to repulse neighboring particles, forming a secondary charge layer (Stern layer).

Covering the surface of particles with a stabilizing polymer, as described above, is, however, undesirable if the original particle surface is to be used subsequently to participate in reactive functionality, as the surface reactivity is then strongly suppressed by the presence of the polymer, together with the effective surface area being reduced by the polymer occupying space in the surface microstructure. Furthermore, the colloidal floc or aggregate structure which develops in a colloidally unstable suspension due to the absence of dispersant, can provide positive increased porosity, and, crucially, increased pore connectivity, in the final coating layer, thus enhancing the permeability of the coating within its pore network structure. Permeability to fluids allows rapid transport within the porous medium, driven either by wetting and capillarity for liquids during absorption, or pressure and surface adsorption potential leading to concentration diffusion-driving gradients for gases.

The references (Gane et al., 2020; Gane et al., 2021) [4,5], highlight a particular case illustrating the advantages of increased surface area access, and the opportunity to develop surface pressure gradients, to enhance the sorption on, and reactivity of gas with, the surface of a mineral particulate coating. In this case the increased performance of a surface flow filter, coating fine particulate calcium carbonate designed to react with atmospheric pollutant NO*_x_*, is achieved by applying the coating in the form of an array of pixelated dots using a convenient coating method, such as pin coating. The desired reaction with NO*_x_* is to form calcium nitrate (Ca (NO_3_)_2_), the aim being not only to lock-in the polluting gas but to turn the reaction product into a useful material following the principle of circular economy, in this case the nitrate of calcium providing a micronutrient fertilizer [3,4,5]. However, the reaction that is desired generates the in-situ release of CO_2_, which naturally, though small in quantity, is not desirable in the fight against greenhouse gas emissions, Equation (1).
CaCO_3_ + H_2_O + 2NO*_x_* _= 1,2_ + ½ O_2_ → Ca(NO*_y_* _= 2,3_)_2_ + H_2_CO_3_(1)
H_2_CO_3_ → H_2_O + CO_2_ ↑

To overcome this downside, inclusion of a mesoporous ancillary mineral has been proposed to capture the CO_2_ [3]. Traditionally, ancillary minerals such as perlite and sepiolite have been used as inorganic ingredients, such as in soil and in construction, forming artificial muds to enhance cohesion due to their high surface area, and, in the case of sepiolite, due its fibrous morphology. Additionally, sepiolite is considered a suitable additive to livestock food, for example to improve poultry egg quality [6]. The fibrous structure of such silicate-based clay ancillary minerals enables adsorption and retention of water and various other, often targeted, substances, such as toxic and undesirable impurities from liquids and air. In rheological terms, such minerals form hydrogels with water having high viscosity and long viscoelastic recovery time after shear. Therefore, the rheological behavior of suspension coatings that contain such minerals affects the end-point coating structure, including pore morphology, uniformity, and porosity, which, in turn, influence the surface sorption properties. Furthermore, due to the high surface area and/or aspect ratio of such ancillary minerals in the coating color matrix, once dried, and depending on orientation, they exhibit extensive volume fraction occupancy in the coating, and this feature can aid coverage of rough substrate surfaces [7,8].

Addressing the rheological aspects of the current example case of a challenging functional coating application, we have studied the stress-related viscoelastic and dilatant structure behavior of the complex material suspension used to form the reactive coating layer using shear, oscillation, and extensional rheometric methods. We report the findings, which allow us to differentiate between the role played by the ancillary mineral in respect to surface water immobilization and/or internal absorption versus that of its physical particle size and morphology. In addition, the inclusion of micro nanofibrillated cellulose (MNFC), a critical component needed to maintain moisture (and binding) in the coating to support the reaction in Equation (1), is shown to reduce extensional strain-rate induced dilatancy, and, in so doing, enables the necessary rheological properties to be achieved so as to support the application of the required uniform pixelated coating.

## 2. Materials and Methods

### 2.1. Mineral Particles

The ground calcium carbonate (GCC), used as the NO*_x_* reacting component in the example coating construct studied here (Equation (1)), was produced from Norwegian marble, wet ground chemical free (Omya Hustadmarmor AS, Molde, Norway). It has a nominally individual dispersed particle size distribution of 60 *w*/*w*% < 1 μm and 95 *w*/*w*% < 2 μm, with a zeta potential of ζ = −27.13 mV at pH 9.2. It is used here in its dispersant-free form to provide retained surface reactivity, and, thus, is extremely pure and essentially process-chemical-free. Figure 1a shows the flocculated/aggregated nature of the GCC in its natural undispersed (dispersant-free) state, rendering its effective particle size larger than in the above-reported dispersed state.

The particle size of the GCC in the required undispersed (as delivered) state was measured using static laser light diffraction intensity from dilute aqueous suspension (Malvern Mastersizer, Malvern Instruments, Malvern, UK) as a function of the scattering angle, providing an equivalent light scattering diameter, *d*, which is cumulatively distributed over the particulate sample volume and reported as the volume percent (*v*/*v*%) of the material particles smaller than or equal to *d*, the values of which are shown in Table 1.

Two example ancillary minerals are considered for the capture of in-situ generated CO_2_ (Equation (1)).
(i).fused milled perlite, a naturally occurring volcanic glass (potassium aluminum silicate), which, when heat expanded and milled, has an eggshell-like platy morphology (Omyasphere^®^ 200, Omya International AG, Oftringen, Switzerland), having 98 *w*/*w*% of particles ≤ 200 μm [8,9], Figure 1b, and(ii).an intercalating mineral, sepiolite (magnesium silicate hexahydrate, palygorskite) Figure 1c, a soft white clay having a needle-like fibrous particle morphology, displaying particles of 1–5 μm in length [10], historically known for the manufacture of clay smoking pipes [11].

In the case of fused perlite, comprising ultrafine pores, CO_2_ gas condensation occurs under confinement within the pore network following the Joule-Thomson (Lord Kelvin) effect, whereby a confined gas condenses to a liquid at a chemical potential below that corresponding to liquid–vapor coexistence in the bulk. In contrast, sepiolite effects the capture of CO_2_ by adsorption via the function of surface –OH groups leading to acid-base intercalation, Equation (2).
−(OH^−^)_2_ + CO_2_ → −CO_3_^2−^ + H_2_O(2)

### 2.2. Formulating the Coating Suspensions and Coating Colors

Advantageously for the reaction in Equation (1), a controlled moisture level can be held in the coating by using micro nanofibrillated cellulose (MNFC), which provides a multifunctional role, namely humectant, binder, and water retention aid all in one [12]. The MNFC is conveniently made by mechanical disintegration of the same fiber source as is used for the coating substrate, namely ‘over-recycled’ newsprint—see section on substrate production below. The presence of MNFC in the aqueous coating suspension formulation contributes to a workable gel-like water-retaining consistency [13].

Mineral blends between GCC and each of the ancillary minerals, in turn, were formed to constitute the inorganic reactive part of the coating formulation. All formulations derived from GCC, perlite or sepiolite alone, or with partial replacement of GCC by perlite or sepiolite, were mixed in their respective ratios to constitute in total 100 parts by weight (pph) of mineral particulate content in each suspension. Two ancillary mineral addition level ratios were used, namely, 95:5 pph and 90:10 pph by weight GCC:perlite/sepiolite, Table 2 and Figure 2. The respective coating colors in which MNFC was added were made with 10 pph MNFC in relation to the 100 pph mineral content, also shown in Table 2. Solids content ranged from 50 *w*/*w*%, established to be suitable for GCC only, to 25 *w*/*w*%, adjusted to overcome the highly gel-like behavior and viscoelasticity arising from the inclusion of the ancillary minerals. Mixing was done under high shear using a Diaf mixer (Pilvad Diaf A/S, Fredensborg, Denmark).

It is to be expected that the nature of a specific application will demand an optimal use of the additive flow modifier chosen, such as MNFC. The focus of the work described in the paper is to illustrate the development of a formulation containing undispersed mineral mixtures for the example application of pin coating onto a substrate to form a surface-flow gas filter. The application is, therefore, specific, in that the capture of the NO_x_ target gas on calcium carbonate, whilst simultaneously arresting the release of the in-situ formed CO_2_, demanded a defined triple action of water retention, binding, and humectant effect derived from the MNFC. Though limiting any exploration of the rheological effect over a range of addition levels for MNFC, it was found in practice that less than 10 pph MNFC simply resulted in insufficient binding to the highly porous substrate, as well as being insufficient to retain the necessary moisture to drive the desired reaction in Equation (1) under possibly low humidity conditions. For these reasons the illustrated study is confined to the chosen level of 10 pph MNFC.

### 2.3. Substrate and MNFC Preparation

The substrate designed for the filter application is based on forming a paper-like sheet from ‘over-recycled’ cellulose fiber, i.e., paper and board waste in which the fiber content has already been weakened due to manifold previous recycling, using a laboratory sheet former with wet pressing [3]. The fiber source we chose was newsprint, Figure 2a, and the substrate for the filter formed from it, after further refining (in this context referred to as ‘over-recycled’ weakened cellulose fibers), is shown in Figure 2d.

To produce the MNFC, the water soaked, and disintegrated pulp suspension of the multi-recycled newsprint was fed through a super mass colloider (Masuko, Kawaguchi, Japan) comprising a rotating abrasive grinding stone (1500 min^−1^ (rpm)) in contact with a static stone of similar material, Figure 2b [14,15,16]. Hydrodynamic pressure and shearing forces generated by the grinding stones resulted, after seven cycles, in final breakdown into nanofibrils branching from the parent wall, hence the term micro nanofibrillated cellulose (MNFC). Typical MNFC fibril morphology, together with example length and width dimensions, are visible in detail in Figure 2c.

### 2.4. Coating Application: Pin Coating

The formation of the 2D pixel array of coating by depositing and withdrawing pins loaded with the coating color suspension places the coating color under high strain and extension, Figure 3, followed by filament formation and breakage upon retraction of the pin array [3,17]. The weakly bonding fibers used for the filter substrate form a highly absorbent porous medium, and so good water retention in the coating color applied to the surface is important to prevent a too-rapid dewatering, in order to avoid extremely high dilatancy during retraction of the pins [15,16]. We, therefore, made the comparison between minerals in water suspension only with similar suspensions in which MNFC was included, acting as a gellant through fibril flocculation [18,19]. The additional water trapping and release properties from within the gel-like material readily support the formation of an outer water-rich boundary [20,21,22].

As described earlier, coating suspensions were formulated starting with GCC alone, initially at a solids content traditionally used for GCC containing paper coatings, but in this case without the addition of a stabilizing dispersant [23,24,25,26,27,28]. Further formulations were then constructed by step-wise addition of the gas-sorbing minerals perlite and sepiolite, respectively, under vigorous mixing to ensure the greatest possible deagglomeration, Table 2 [25,26].

Three main controlling criteria prevail over the end-use and processability of the studied coatings. Firstly, maximum porosity and permeability to enhance gas pick-up via flow and diffusion should be reached. Secondly, the effective suspension particle size of the materials is large: the GCC is flocculated, the perlite has large structural shell-like particles, and sepiolite has high aspect ratio particles sweeping out large volume occupancy under disordered alignment in suspension [29]. Thirdly, the ancillary minerals are porous and absorb water (sepiolite also binds water to its surface), thus displaying effectively lower suspension solids density than GCC for the same weight fraction addition. This latter effect additionally increases the effective volume fraction, resulting in rheological properties associated with an otherwise higher solids content [30].

### 2.5. Rheology of the Suspensions

The application of undispersed mineral particles is rheologically challenging due to re-agglomeration of the minerals after cessation of vigorous mixing. Sepiolite itself forms a gel-like structure due to the surface immobilized water bridging between particles. Therefore, when such minerals are introduced into suspension, the volume of the matrix occupied by rotation of particles during shear increases, giving rise to non-linear viscoelastic behavior, with high yield stress and dilatant response at high shear [31,32]. Such rheological properties, otherwise unsuitable for continuous flow metering coating application methods, can, however, be favorable in the context of high load pixelated coating application, provided, of course, they are suitably controlled [3,4,5]. It is, therefore, essential to understand the full rheological behavior of the coating colors in order to ensure processability, and to define new criteria for runnability in such emerging functional coatings and their related, often more exotic, coating application methods.

Pin coating application, as such, requires neither high coating speeds nor high contact forces, both of which would eventually break the weak recycled fiber-fiber bonds of the substrate. Nonetheless, viscoelastic and low shear rate properties of the coating color are important for precise pick-up of the coating suspension onto the substrate, depending on the transfer from the pin tip during withdrawal after initial contact with the substrate. Both good sliding of coating along the stem of the pin to refresh the tip in contact with the substrate, and final separation from the tip of the pin can be achieved if the rheological properties are correctly tuned, (i) to approach controlled linear viscoelastic behavior, (ii) to minimize rheopectic and dilatant behavior, (iii) maintain low but controlled extensional viscosity, and (iv) develop sufficient water retention properties in the coating to allow pin withdrawal whilst promoting rapid immobilization of the color once the coating has been deposited, i.e., a condition of minimum sagging tendency.

Before embarking on a suitable rheological study of the functional mineral suspensions we undertook a careful overview of the process that needs to be replicated as closely as possible using readily available rheology methods. The process, as mentioned above, consists of a number of steps. We break these down to establish an understanding of their individual boundary conditions for the suspension flow behavior.

(a)Loading the pin coater array

This is a low shear procedure consisting of dipping the pin array into a bath or reservoir of the coating suspension. Due to the flocculated nature of the suspension sedimentation is prevented inside the reservoir, and only gentle stirring is necessary to prevent expulsion of water from between the flocs, which otherwise would form a surface layer, especially in the case of the minerals alone without introduction of the MNFC. Effectively, therefore, the mineral suspension is initially close to the rest state but without any water separation effect. The pin loading process step will still be dominated by the state of flocculation of the mineral particles in their unstable state of dispersion. This will mean that a weak yield stress followed by a partial breakdown of flocs is involved. In the contrasting case where the gel-forming MNFC is present, the mineral flocculation will be accompanied by the gellant nature of the MNFC suspension when at rest. The low shear process then breaks the static gel followed by thinning of the gel-dominated rest state.

The conclusion, as to the governing forces dominating overall in this first stage (i) action of loading the pins, is more one of surface tension and wettability of the pins as the flow driver, whilst the floc structure and resistance to flow at low shear is the brake. The resulting balance in favor of wetting is sufficient for suspension uptake onto the pins and retention once loaded by both adhesion and cohesion. Response to low strain and low shear rate conditions, in respect to monitoring thinning, is sufficient to study this step rheologically. (The role of surface tension is again visited during the measurement of extensional response.)

(b)Transfer to the substrate

Unlike a full study of the rheological material properties per se, the aim here is to have simply a relevant investigation of the coating flow boundary conditions at contact with and transfer to the substrate surface. If the substrate were impermeable and not absorptive, then a wide range of study would be needed to reveal flow onto the substrate and subsequent spreading and settling, including longer term relaxation effects extending into the following step (iii) of filament separation and release. Re-establishment of equilibrium conditions, i.e., change in viscosity under shear over time, for example, time dependent increase in viscosity during relaxation to the rest state upon cessation of shear, are not relevant for one dominating reason. That reason is; the substrate is permeable and highly absorptive, especially arising from the use of over-recycled fiber, which becomes easily wetted and itself absorbs water readily. This means that the process is an extreme case of paper coating onto a highly absorbent substrate. In addition to the strong absorption, the coating condition having flocculated mineral suspension results in a highly permeable structure in the suspension, having cavities within a particle matrix, through which rapid dewatering can occur. This combination of structurally permeable and wetting properties, leads to almost instantaneous immobilization of the coating as solids content rapidly rises on contact with the substrate. Immobilization is, therefore, more rapid than long timescale rheological structural relaxation in the coating itself, such that the coating retains the internal structure associated with the applied wet state flocculation and agglomeration. Thus, it is more important to consider how higher solids content due to dewatering manifests itself as apparent rheopexy and dilatancy. In the case of MNFC addition, the water retention properties are enhanced and the immobilization rate is somewhat slower (Dimic-Misic et al., 2013) [6], but the transition to step (iii) is faster than the time required for relaxation equilibrium to be reached, and so even in this gel-like case, studies of rheological change over time remain largely irrelevant.

(c)Withdrawal of pins and filament separation

Under the withdrawal of the pins, the semi-immobilized dewatered coating, in the form of a pixel dot on the substrate surface, becomes rapidly extended to form a filament prior to separation. The response to rapid extension is the initial key property here. This can be explored initially using oscillatory rheometry to study the early stage quasi linear elastic region as strain is applied. This is then followed by two other techniques, exploration of dilatancy under high shear and induced fracture brittleness through strong particle-particle interaction resisting rapid changes in strain, i.e., high frequency deformation. Both of these analyses also apply when MNFC is present due to the gel structure it creates. Firstly, the gel aids flow due to its thinning behavior, i.e., thinning as strain and shear rate increase. However, secondly, gel hardening cannot be overlooked, and once again high frequency analysis aids prediction of this. In both latter cases, spanning the oscillatory rate from low to high maps the first breakdown of the quasi linear viscoelastic region, leading to reduced elasticity and viscosity as thinning takes over after structure breakdown, followed by a secondary increase in both elastic and viscous response as either high frequency induced agglomeration and/or gel hardening occurs depending on the system under study, i.e., without versus with MNFC, respectively. To confirm whether homogeneous filament separation can be achieved, extensional effects are studied directly using a filament drawing method, i.e., effects which clearly differentiate between the brittle breakdown of the filament under the pinch effect of surface tension in the case of mineral suspension alone versus the gel stretching and lubrication effect contributed by MNFC nano-fibril water retention coupled with microfibril orientation.

Given the descriptive process steps above, the process-relevant rheological study as described is reported here employing the following instrumental measurement techniques.

### 2.6. Viscoelastic Behavior

The ancillary minerals exhibit a strong effect on the rheology of the suspension, relating to their individual morphology and lower particle density compared with GCC, hence exhibiting high volume concentration per unit weight in the formulation, especially under shear [7,26]. In considering the properties defined above ((i)–(iv)) we studied the stress-related viscoelastic and dilatant structure behavior of the complex material suspension.

The viscoelastic rheological investigations were performed at 23 °C by using an Anton Paar 300 (Anton Paar Austria GmbH, Graz, Austria) oscillatory shear rheometer with plate-plate geometry PP-25. The modifying properties of the water retaining MNFC fibrils, is found to help in creating a more uniform rheological response from the mixes of dispersant-free GCC [31]. However, in respect to the study of shear flow, as observed previously with similar materials, plate-plate geometry is prone to apparent wall-slip and shear inhomogeneities, in particular for such complex shear thinning coating colors containing MNFC [33,34].

To allow for this potential for apparent wall-slip with solids-depleted boundary layers, two different plate gaps were used in respect to the mineral particle morphology and suspension viscosity: for the coarse non-dispersed carbonate coatings (100 pph GCC) the gap between the plates was 1 mm, while for formulations of pure ancillary minerals, due to high gelation and platelet or needle morphology (100 pph perlite and 100 pph sepiolite), the gap was increased to 1.5 mm [34]. This practice of changing the gap between differing samples in the avoidance of some undesired behavior, such as solids depletion, can raise other severe difficulties in terms of comparability of data. To a first approximation, linear inverse dependence of shear on increasing plate gap is assumed here. For complex systems of disparate particles in highly viscoelastic systems the assumption is extremely dubious, and, if possible, is to be avoided. We are, nonetheless, here forced to accept this approximation out of practicality. In addition, to avoid any memory effect in the sample, derived from prior deformations, all samples were pre-sheared at an angular frequency of 10 (rad) s^−1^ and strain deformation *γ* of 0.1% for 10 min, followed by a rest stationary state time of 10 min [19,20].

Complex viscosity (*η**), and dynamic moduli (storage modulus (*G′*) and loss modulus (*G″*)), were measured as a function of angular frequency (*ω* = 0.1–100 (rad) s^−1^) using oscillatory tests. A frequency sweep test was performed within the quasi linear viscoelastic (LVE) range of the samples, which was previously determined adopting an amplitude (strain) sweep test using constant angular frequency (*ω* = 0.1 (rad) s^−1^) with varying strain amplitude over a range of *γ* = 0.01–500%. A detailed description of oscillation measurements was presented in earlier work [6,12,17].

### 2.7. Yield Stress

The apparent effective yield stress depends on the determination method and the flow geometry as well as potential apparent wall slip in gel-like materials, which is a major factor adding to the difficulties of yield stress estimation for gel-like MNFC suspensions [7,20,21]. For oscillatory measurements, the maximum in the elastic stress (*τ*_s_) corresponds to the static elastic yield stress (*τ*_s_^0^), which is determined as the first point of deviation from the quasi LVE, corresponding to the critical strain value (*γ*_c_) such that
(3)$$\tau_{s}^{0}=G^{\prime }\gamma_{c}$$

### 2.8. Shear and Strain Response

When determining the response to shear the factor of large particle disruption in the flow is extremely important and becomes crucial in the pin coating process in question [35].

Flow curves were initially measured as a function of decreasing shear rate, spanning a wide shear rate range ($\dot{\gamma }$ = 1000–0.01 s^−1^) with a logarithmic spread of data point collection, and duration ranging from 1 to 100 s to attain equilibrium. This initial procedure eliminates the question of static yield stress when the shear rate regime is then subsequently reversed, as occurs when retraction of the pin coater is initiated. Instead, the dynamic yield stress becomes relevant. The result from the shear rate study is firstly the shear thinning of an already sheared system, as would be the case under continuous reservoir stirring, and whether there is presence of dilatancy as shear-induced agglomeration occurs. Thixotropy, shear thinning effect as a response to zero or low-shear rate, developing over time, and a typical rheological property of gels, is, therefore, not investigated here, since time to thin is not relevant as the system is already broken down prior to the shear rate study. Shear thinning as a response to rate of strain change is the more relevant discussion as the pins are removed and the coating must separate, and this is advanced by the understanding derived from the high frequency measurements discussed above. Beyond the dynamic yield stress (*τ_d_*^0^), the rheological property of the suspension is considered to be in the fully flowing condition. In the case of coating color suspensions, we took stress data from the upward (increasing shear rate) steady state flow curves ($\dot{\gamma \;}$= 0.01–1000 s^−1^) applying a data point measurement duration of 10 s [16,18]. Therefore, for *τ*_d_^0^ determination data from steady state curves applying the Herschel-Bulkley equation (Equation (4)) were used, according to
(4)$$\tau_{d}=\tau_{d}^{0}+k{\dot{\gamma }}^{n}$$
where *τ*_d_ is dynamic shear stress, *τ*_d_^0^ is apparent dynamic yield stress, *k* is a constant, termed consistency, and $\dot{\gamma \;}$ is shear rate.

The Ostwald-de Waele expression was used to fit the thixotropic shear thinning flow response of the dynamic viscosity (*η*) from the steady state flow curves as a function of shear rate ($\dot{\gamma })$,
(5)$$\eta =k{\dot{\gamma }}^{n-1}$$
where *k* and *n* are the consistency and power-law index, *n* = 0 indicates a Newtonian fluid and *n* > 0 pseudo-plastic (shear thinning) behavior [6].

Complex viscosity behavior (*η**) as a function of angular frequency was evaluated from frequency sweep measurements, as described in the previous section on viscoelastic analysis. Here, an equivalent power law according to the Ostwald-de Waele empirical model was fitted to the experimental data also for the complex viscosities (Equation (6)), allowing for a comparison of the power law for complex viscosity (*η**),
(6)$$\eta^{*}=k_{c}{\dot{\gamma }}^{n_{c}-1}$$
where *k*_c_ and *n*_c_ are the equivalent consistency and power law index for the complex viscosity, respectively. Similarly, *n*_c_ = 0 indicates a Newtonian fluid and *n*_c_ > 0 pseudo-plastic (shear thinning) behavior.

### 2.9. Elongation

Amongst the methods used to determine the extensional rheology of viscoelastic samples, two types of rheometers are commonly applied, which are conceptually similar. One adopts filament stretching (FiSER) and the other capillary break-up (CaBER) [35,36]. In both cases uniaxial extensional flow is applied to the sample bounded between two rigid plates that when in separation force the liquid to create a bridge [37]. In the filament stretching case, as exemplified in this work on a Haake CaBER^TM^ (Thermo Fisher Scientific, Waltham, MA, USA), the plates are drawn apart actively using a defined rate profile until the filament breaks [31,38]. However, depending on the nature of the material, the way in which the sample is stretched differs. In the capillary break (CaBER) case the cylindrical endplates are separated either linearly or exponentially applying a sudden step-stretch which is then held at a fixed axial separation and the capillary observed until break-up occurs [31]. The strike time for the CaBER plate separation is noted as 50 ms, which is the value defined in the control software. However, when looking at the diameter profile this does not seem correct in practice. We consider that the actual strike time is likely to be shorter, basically starting once the filament diameter decreases, in some cases after a small initial increase.

The filament diameter (*D*(*t*)) is monitored throughout the whole experiment, including both during the plate separation and after cessation of the separation, using a laser illuminating beam mounted at the same height as the filament mid-point [35,38]. As a result, a liquid filament of axially symmetrical geometry is formed, Figure 4, which, for the case of a purely viscous material displays the expected “pinched” effect caused by reducing diameter. Once the top plate is halted, the filament undergoes progressive thinning driven by surface tension. This aspect is critical when the suspension contains large particles, since, ideally, this thinning results in capillary pressure flow in the suspending liquid phase being resisted by the induced filament internal resistive stress [36,38]. The behavior of the stress balance with capillary pressure is the key to understanding the elastic component effect in viscoelastic systems under elongation. However, this can be disrupted if the particle size in suspension is large in comparison with the filament diameter [39]. Nonetheless, even if this happens, the experiment is relevant to the process studied here, as in practice the filament will be forced to break by the presence of such particles unless the continuity of the water meniscus around the particle can be maintained [31].

Assuming an axially symmetric shape of the filament, the Hencky strain *ε*_H_ is given by
(7)$$\varepsilon_{H}\left(t\right)=2\ln\left[\frac{D_{0}}{D\left(t\right)}\right]$$
with *D*_0_ being the initial filament diameter and *D*(*t*) being the measured diameter at time *t*. The strain rate ${\dot{\varepsilon }}_{H}$ [39] is, therefore,
(8)$${\dot{\varepsilon }}_{H}=-2\frac{\mathrm{dln}D\left(t\right)}{dt}$$

For viscoelastic systems, as represented in Figure 4b, the elastic stress grows as the strain increases and eventually it dominates over the viscous stress from the liquid at the onset of the elasto-capillary thinning regime. In the case of particulate suspensions, if the system is strongly elastic, as is the case in this study, and potentially within the system there are two or more elastic structures, then the interaction forces between the particles can be existing in a series of states, each state ranking from weak to stronger, and each having a progressively more rapid equilibrium recovery constant than the surface-tension-based capillary pressure relaxation.

In this regime, the decay of the mid-plane diameter of the filament is exponential and can be expressed as
(9)D(t)D0≈(G′D04σ)13e−t/3λE
where *G*′ is the elastic modulus, *D*_0_ the filament diameter at zero time, σ is the surface tension, *t* the time, and *λ*_E_ the relaxation time of the largest aggregate. In practice it was shown that the elasto-capillary thinning regime, within which the filament thinning is counteracted by the elastic stress components only, is observed only at very high strains and thus at a late stage of the measurement close to filament break-up. Therefore, as previously observed with mineral pigment-cellulose fibrillar complex suspensions, it is not possible to identify the elasto-capillary thinning regime due to very fast break up of the filaments and thus relaxation times cannot be evaluated in such a case [31].

Extensional flow of particulate-MNFC suspensions follows that of a flocculated paste system in which agglomerates of different network elasticity that continuously resolve hierarchically within step-wise breakdowns, that initiates from a weakest towards the most elastic flocs. Extensional viscosity is observed to be a function of increased strain and highly correlated, therefore, to the elastic modules (*G*′) of the suspension [32].

For the CaBER experiments, 6 mm diameter plates were used with the sample loaded at an initial plate separation distance of 2 mm. The plates were then moved apart following a linear profile with a strike (step-stretch) time of 50 ms to a final distance of *L*_max_ = 9 mm. All measurements were performed at room temperature.

## 3. Results and Discussion

SEM micrographs of mixtures of calcium carbonate with ancillary minerals are shown in Figure 5. The partial replacement of GCC develops an increase in packing volume (lower packing density), particularly with platy ancillary mineral particles, as shown in the case of partial replacement of GCC with 5 pph perlite, as presented in Figure 5a. Small aggregates of perlite are found dispersed within the GCC matrix making the structure heterogeneous. With the addition of further perlite (10 pph), the presence of aggregates between GCC particles is increased, and the perlite particles protrude above the surface of the mix, as shown in Figure 5b.

Acicular aggregates present upon substitution of GCC with 5 pph of fibrous sepiolite indicates a good dispersion of GCC within the hair-like rods of sepiolite with discrete aggregates within the empty regions formed between the edges of flocculated calcium carbonate domains, Figure 5c. At a higher substitution amount, protrusion of sepiolite particles outside the compact surface consisting of GCC particles also becomes prevalent, Figure 5d.

Upon the addition of fibrillar MNFC as binder and flow modifier, forced collision between the components of the mix upon shearing acts to disperse plate-like perlite and fibrous sepiolite, giving rise to a more homogeneously distributed coating structure, Figure 6. Since sepiolite (hydrated magnesium silicate) has a microcrystalline morphology consisting of fiber-like ribbons and cavities it can coalesce with the more isometric round shaped carbonate and fibrillary MNFC to give a somewhat consolidated combined homogeneous mix structure, which clearly contrasts with the disruptive plate-like perlite packing effect leading to clear material separation according to size and shape—compare Figure 6a,b, showing the disruptive packing caused by the ancillary perlite, with that of Figure 6c,d, showing the more homogeneous packing in the sepiolite case.

### 3.1. Rheological Observations

Due to the different packing morphologies of the ancillary mineral particles, together with their respective colloidal forces in the mixtures of GCC, the viscoelastic moduli parameters *G*′ and *G″* have a single linear response to applied strain within the quasi linear viscoelastic (LVE) region. Figure 7a shows the viscoelastic moduli at 25 *w*/*w*% solids content in the various mixed mineral suspensions when no flow modifier is present. As the proportion of ancillary mineral is increased at constant solids content so the magnitude of the moduli increases.

The key factor for the increase in viscoelasticity arising from the ancillary minerals themselves is reflected in the static strain within the viscoelastic region, which increases as a function of applied strain for the suspensions of the pure ancillary minerals alone, Figure 7b. By also comparing the solids content of each ancillary mineral alone in Figure 7b, the volume fraction occupied in the static randomly oriented semi-locked structure increases dramatically for the cases of the poorly packing platelets and needles, respectively, as solids content is increased from 25 *w*/*w*% to 50 *w*/*w*%. The eventual breakage of the elastic floc structure allows the particles to orient in the direction of strain, and the modal collapse of the viscoelastic parameters seen at the higher solids content reflects this orientational effect, Figure 7b. At even higher strain, the aligned packing can no longer be maintained as physical particle jamming occurs, leading to dilatancy, typical of strain induced aggregation of such undispersed materials.

The study of stress response to applied strain at increasing solids content is repeated for the case of the mineral mixes, as shown in Figure 7c, again without addition of flow modifier. Due to the high-volume occupancy related to the ancillary mineral clusters, even within the GCC host matrix, the jamming effect seen at 50 *w*/*w*% also occurs as GCC is partially substituted with perlite and sepiolite. The effect is so strong that keeping the system within a desirable static stress window matching a GCC suspension alone of 50 *w*/*w*%, i.e., similar static yield stress (*τ*_s_^0^), requires significant dilution.

### 3.2. Effect of MNFC Addition

In contrast, oscillatory measurements of coating colors with mineral mixes plus additional flow modifier MNFC, Figure 7d,e, show structuration more gradually arising from material interaction as a result of increasing solids content from 25 *w*/*w*% towards 50 *w*/*w*%. Once again these interactions are seen via the developing quasi linear viscoelastic (LVE) behavior, seen as storage (*G*′) and loss (*G″*) moduli dependence on strain within the range *γ* = 0.1–1%.

The suspension of minerals, although studied rheologically by tradition as a liquid slurry, is more like a hybrid consisting of an agglomerated particle system lubricated by water. Thus, viscoelasticity is present with and without MNFC. In the first instance without MNFC the elastic component derives solely from particle-particle interactions and the expulsion of lubricating water from between particles due to flocculation and shear induced agglomeration. In the case when MNFC is present, the gel-like nature shares with this hybrid form, leading to elastic gel-breakdown displaying thinning and somewhat retained lubrication due to the strong water attractive nature of the cellulose nanofibrils attached to the parent orientable microfibrils. Thus, both cases exhibit viscoelasticity but of a different nature. Dual effects in the case of added MNFC are likely at play, namely the flocculation of non-stabilized dispersant-free mineral components, as discussed above and seen in Figure 7a–c, particularly for the high aspect ratio ancillary minerals in combination with the GCC, and additionally the gel-forming behavior of the MNFC itself, respectively. Both flocculated systems and gel-like systems display yield effects, but derived from different mechanisms, as explained for viscoelasticity above, i.e., strong flocculated particle-particle interactions and water expulsion from between the particles versus the water immobilizing effect of trapped water between the water-binding nanofribrils. The yield is thus also derived differently, namely floc breakdown alone versus floc and gel breakage together. The strong viscoelastic behavior interestingly displays this dual behavior as a bimodal rheological response consisting of two almost distinct LVE regions, Figure 7d,e. These can be designated by two distinct elastic moduli, *G*_1_′ and *G*_2_′, which after the initial increase under strain then decrease separately after reaching two discrete, respective, critical strains (*γ*_c1_ < *γ*_c2_). The bimodality is more pronounced for sepiolite containing coatings than for perlite, as can also be seen in the non-linear behavior in loss modulus (*G″*), Figure 7d. The amplitude sweep measurements, Figure 7e, show strong viscoelastic behavior and higher static stress *τ*_s_ for the formulation with GCC mixtures with the added ancillary minerals. The effect is strongly solids content dependent, and, naturally, even stronger at the higher amount of ancillary mineral, both sepiolite and perlite. By way of comparison, the static yield point, defined as the maximum of static stress, for the higher solid content 50 *w*/*w*% of 100 pph GCC was still lower than the significantly greater values for the ancillary mineral containing coatings even at the considerably lower 25 *w*/*w*% solids content.

Critical strain values (*γ*_c_), at which *G*′ decreases, are greater for larger agglomerate sizes, and, as mentioned previously, show the bimodal decrease in *G*′ for both perlite and sepiolite, indicating the presence of multimodal orientational phenomena, as will also be revealed when reporting the extensional behavior. In addition, the frequency sweep measurements presented in Figure 7d,e show the typical behavior of gel-like properties for the MNFC containing GCC coatings, now having much more pronounced multimodal structuration when ancillary particles are in suspension. The water-binding properties of sepiolite, in particular, influence more strongly the rheological behavior. This is seen as creating very high magnitude *G*′ that remains independent of oscillation frequency (*ω*) when strain is set to be less than the first critical strain point (*γ* < *γ*_c1_). In these cases, their multimodal nature is revealed through non-linear *G″* behavior at low frequencies.

When MNFC is additionally present in suspension, its effect of acting to disperse agglomerates by increased collision probability, results in lower values for *G*′ and *G*″, together with the water-trapping gellant properties associated with MNFC itself, Figure 8a,c,d. Under rapid change in shear rate, due to the retention of the lubrication effect and interference of shear induced particle-particle agglomeration, the presence of MNFC reduces induced dilatancy. This is better described in relation to the process dynamic of separation, however, as a reduced observed brittleness under rapid strain application. As for gel-like materials in general, the response of *G*′ and *G″* for coatings that have interacting aggregates in the absence of flow modifier, for either single component minerals or the respective GCC plus ancillary mineral mixes, is highly dependent on angular frequency (*ω*), increasing for higher frequencies due to the gel hardening response. We see that this effect is more pronounced for sepiolite than perlite containing suspensions, being also strongly solids dependent, Figure 8a,b [25,26]. When suspensions contain only a single mineral, then the influence of the particles is distinguishable in response to static floc stretching and hardening with higher frequencies in the cases of 100 pph perlite and 100 pph sepiolite. Trapping of water in cavities between the particles of perlite and within and between needles of sepiolite is also demonstrated by higher values of elastic moduli when compared with the case for 100 pph GCC suspension at the same solids content.

A very valuable conclusion can be drawn from the data in Figure 8. Starting from low frequencies, both *G*′ and *G″* first decrease and, after reaching a local minimum, then both increase at higher frequencies. This behavior breaks the Kramers-Kronig relationships between *G*′ and *G*″. The Kramers–Kronig relations are mathematical relations, and apply to the specific analytical case of complex numbers where either the real part can be derived from the imaginary part (or vice versa) of response functions in physical systems, and here is the point to stress, that are stable. This means that they only apply in cases where the causality, i.e., driving condition of the response, is stable. This stability is required to imply the condition of the Kramers-Kronig relationship analysis, and, conversely, an implication from the analysis that the causality belongs to a stable physical system. Physical examples of such stable causality are discussed via Kronig on the theory of dispersion of X-rays in a static environment system (Kronig, 1926) and Kramers discussion of the diffusion of illuminant by atoms (Kramers, 1927) [40,41]. In our rheological analysis, the system is anything but predictively stable, unlike traditional fluid samples used to exemplify standard equilibrium rheological measurements, and transitions in our systems occur between completely different interactive states. This means that Kramers-Kronig’s inter-component predictive power for complex rheological moduli can only be applied over segments of the response corresponding to just one physically stable system applicable in each segment separate from the systems prior and post. Thus, a case, such as exemplified here, of shear induced effects on particle-particle stability/instability will not obey any continuous Kramers-Kronig relation. Wording it another way, the fact that the Kramers-Kronig relation is broken shows that the system is not a physically stable one but is one consisting of process-consequential physical transitions, i.e., precisely the topic of this paper.

### 3.3. Shear Thinning Response

The shear thinning behavior of water-based mineral suspensions, without any flow modifier, within the viscoelastic region under oscillation was traced using the complex viscosity (*η**) response to increased angular frequency (*ω*) under constant strain, as presented in Figure 9a, and as dynamic viscosity (*η*) response at steady state measurements of increasing shear rate ($\dot{\gamma }$), Figure 9b. Shear thinning behavior for both *η** and *η* at high stress changes rapidly to dilatant behavior, which is typical for undispersed mineral particle agglomeration, but the effect gradually decreases with decreasing presence of ancillary mineral, from 10 pph towards 5 pph, in mixture with GCC.

The presence of rigid ancillary pigments and their aggregation within the GCC matrix is effectively shielded with the addition of the high aspect ratio fibrillar MNFC. Both complex viscosity, Figure 9c, and dynamic viscosity, Figure 9d, are displaying more uniformity in the flow curves due to the increased dispersing effect of MNFC. Not all individual spaces between particles within rigid agglomerates are accessible to MNFC fibrils, including regions of immobilized water, and, therefore, there remains a strongly evident dominant influence of sepiolite and perlite on the rheology of the mixed mineral coatings. However, importantly, the ancillary mineral influence no longer extends to inducing dilatant behavior, seen to be absent for both the *η**, Figure 9c, and *η*, Figure 9d, response at high angular frequency and shear rate values, respectively.

### 3.4. Model Curve Fitting (Ostwald-De Waele Parameterization)

The rheological values and model exponential curve fitting parameters obtained with viscoelastic and steady state measurements of pin coating suspensions are presented in Table 3. Complex viscosity (*η**) and dynamic viscosity (*η*) values are reported at the lowest and the highest values of angular frequency (*ω =* 0.1 rad s^−1^ and 100 rad s^−1^) and consequent shear rates ($\dot{\gamma }$ = 0.01 s^−1^ and $\dot{\gamma }$ = 1000 s^−1^) to indicate the enormous difference in initial viscosities relating to the dominating effect of the impact of the addition of the ancillary minerals, not following monotonic power law behavior. The presence of sepiolite and perlite dominate flow behavior for both the quasi LVE domain and under shear with the latter showing more pronounced dilatant behavior at higher shear rates (Table 3). The particle jammed paste-like behavior of the mineral mixes at lower shear rates was also retained for the case when the flow modifier MNFC was added, due to complex interparticle colloidal interactions. However, the presence of MNFC aids greater particle mobility at intermediate and high shear rates, thus reducing shear-induce agglomeration and hence mitigating the dilatant response [42]. For coating suspensions that contain sepiolite as the ancillary mineral, all viscoelastic properties and steady state properties (elastic modulus (*G*′), loss modulus (*G*″), complex viscosity (*η**), and dynamic viscosity (*η*), along with the yield stress, both static (*τ*_s_^0^) and dynamic (*τ*_d_^0^)) increase with the trend to higher packing volume (lower packing density) in the suspension matrix, exhibiting multimodal behavior. In contrast, perlite containing formulations have a more monotonic behavior up to the breakdown of the viscoelastic structure, which may be considered due to its much greater particle size and platelet shape acting somewhat independently of the surrounding finer structure flow.

The difference in response to initial shear, reflected in the fitting values in Table 3, between the wholly flocculated mineral alone suspension versus that of the mixed flocculated and gel-like nature when MNFC is present can be monitored in respect to the effective shear thinning (coefficient n index). Following yield of a flocculated system, the shear thinning occurs initially more rapidly than when gellant is present, due to the induced catastrophic breakage of flocs in the case without MNFC versus the gradual release of held water in the gel formed in the presence of MNFC. The resultant steady state shear viscosity is considered to be a function more of the difference in volume fraction with and without MNFC. The formulations are based on solids content. Minerals have over twice the density of that of cellulose, and so the volume fraction occupancy by minerals is half that of the added MNFC cellulose. Thus, although the weak gel affects shear viscosity less than that of agglomerating minerals per unit volume occupancy, the results experimentally show that the increased volume fraction associated with addition of MNFC leads to a reported higher viscosity under shear.

### 3.5. Rheological Evaluation under Extension

The coating color filament fracture upon application from the pin coater to the base paper controls transfer and uniformity of pixel distribution on the substrate paper. Filament stretching, including internal orientation effects, increases as a function of Hencky strain (*ε*_H_) up to the point of filament breakage, Figure 10a,b. The water adsorption mechanism associated with sepiolite is seen to be detrimental to filament stability, due partly to the resistance to orientation in addition to the normal force effects associated with water binding. The distinct filament breaking point on the CaBER device can be observed through the normalized mid-filament diameter (*D*(*t*)/*D*_0_), where *D*_0_ is the starting diameter, with *D*(*t*) measured at the necking or pinch point as presented in Figure 10c–f.

For the ancillary minerals alone, the stress as a response of stretching rate is, as to be expected, greater, with higher stress for sepiolite than for perlite, Figure 10a,b, relating to the progressive orientation rate of the needle-like particles of sepiolite being slowed at increased solids content. In contrast, for 5 *w*/*w*% and 10 *w*/*w*% mixtures of ancillary minerals with GCC, the filament stretching time to breaking is similar for both sepiolite and perlite addition, Figure 10c,d. The presence of the GCC prevents orientation of high aspect ratio particles and decreases the strain at which break-up of the filament occurs. Therefore, stress and extension of the filament to breakage are properties related primarily to the mineral aspect ratio and orientation ability, ranging from blocky isotropic particles of GCC to egg-shell platiness for perlite and needle/hair-like morphology for sepiolite, which ultimately control the different filament stretching behavior.

Figure 10c,d illustrate the effect of MNFC addition on GCC and ancillary pigments. The addition of the flow modifier acts to decrease the external energy necessary to initiate filament breakage, whilst reducing stress within the filament as a function of the greater rate of extensional strain it can tolerate, which is important for the pin coating application, i.e., it reduces dilatant brittleness (termed shortness).

Comparison between the extensional rheology response of the ancillary mineral sepiolite, which additionally adsorbs water, versus perlite, which primarily absorbs water into a mesoporous structure, is seen in Figure 10a,b, where the more strongly dilatant behavior of sepiolite destabilizes the extended filament earlier.

The dominating property of sepiolite, related to water adsorption, is retained also when MNFC is present, but with mitigated dilatancy upon extension, Figure 10c–f. For ancillary mineral containing coatings in which MNFC disperses rigid needle-like sepiolite and eggshell-like perlite particles within the GCC matrix, the MNFC istelf, as seen in the previous discussion, contributes its own viscoelastic response, arising from its fibrillar structure, manifest in the transient extensional viscosity, *η*_E_^+^, at which break-up occurs. As expected, the *η*_E_^+^ is still highest in the presence of MNFC for sepiolite containing coatings, and lowest for GCC only coatings. For all coatings, the break-up of the filament is solids dependent, as the capillary forces within the filament are larger the denser the packing of the system, and necessarily occurs later the greater the viscoelasticity of the system, Figure 10e.

The data plotted in Figure 10c–f show that the filament pinching during the CaBER experiment is extremely fast, with the break-up happening after just a few milliseconds. In order to calculate the transient extensional viscosity (*η*_E_^+^) it is necessary to separate the diameter data recorded during the separation of the plates from the filament thinning solely driven by capillary surface tension forces. The single phases of the CaBER experiment are more distinctly displayed by considering the Hencky strain-rate, $\dot{\varepsilon }$_H_ = (1/*L*)d*L*(*t*)/d*t*, where *L*(*t*) is the separation of the plates as a function of time. Multimodality of filament extensions and stretching prior to break-up for ancillary minerals, being more pronounced for presence of sepiolite than for perlite, when in the presence of MNFC exhibits the multilevel agglomeration within the matrix, clearly forming hierarchical viscoelastic responses. These hierarchical responses are not decipherable, however, for suspensions without MNFC due to their overall brittleness [42,43,44,45]. The effect is clearly presented in Figure 11, where values of the transient extensional viscosity at the start of extension, *η*^+^_E0_, are shown in ratio to that at the maximum Hencky strain (*ε*_H_), *η*^+^_Emax_, at which filament break-up occurs.

### 3.6. Impact on Coating Performance

The different qualities of pin coating achieved in practice highlight the importance of uniformity in respect to volume transfer per coating dot (pixel). Dilatancy under extension, if present, also brings with it the negative potential for air entrainment associated with brittle fracture of the filament [38]. Before the separation is initiated, the strain is a constant value since the filament length *L* is static prior to elongation during the subsequent removal of the pin from the substrate contact. Removal results in a steep increase in the strain rate, corresponding to the active stretch of the sample filament. Subsequently, the filament thinning, driven by capillary forces, starts and is very distinct, monomodal and sharp for GCC coatings, while having multimodal structuration induced regions prior to final break up for sepiolite and perlite containing coatings as the system becomes brittle but mitigated by the presence of the flow modifier MNFC.

Viscoelastic behavior is a result of colloidal interactions and orientations of the particles in the coating suspensions, which, in turn, govern viscoelastic and flow behavior upon transfer of the suspension to the pin and its separation from the pin during and after application to the substrate. Loading of coating color to the pin is driven by surface tension adhesion to the surface of the pins, as well as the viscoelastic and shear thinning properties of the suspension. The subsequent separation from the pin is driven first by surface tension adhesion of the suspension to the wettable coating substrate surface, followed by filament extension and breakage as the pin array is withdrawn. Ancillary minerals present with their water absorption and/or adsorptive binding properties, together with effective poor packing related to morphology, resulting in high effective volume fraction occupancy in the suspension volume, increase the friction between the fluid layers in a random and unpredictable way [24,32]. As discussed in this work, MNFC as flow modifier improves dispersion, smoothening rheological behavior, and preventing rapid dewatering into the substrate, thus improving coating suspension transfer and uniformity of separation. The coating process, as presented earlier in Figure 3, can be drastically improved by controlling the elongational behavior of the functional coatings being applied, and thus mitigating the limitations of the intrinsic material suspension rheology. As exemplified by the case here of applying a host mineral suspension of GCC containing increasing amounts of ancillary mineral onto a recycled paper substrate, this necessary improved control of viscoelasticity and structuration under elongation can be achieved with addition of MNFC. It is seen to act to transform the process from one of non-uniform transfer and irregular pixel dot size in the presence of 10 pph of sepiolite and perlite as ancillary minerals in GCC suspensions, Figure 12a,b, to that of a more uniform coverage and ease of filament separation. The improvements lead to a more regular array of continuous coating pixels with a greater amount of coating color being transferred to the pixel dot, Figure 12c,d. Height differential between that of the dot pixel and that of the substrate is of great importance for the creation of air flow pressure gradient between the plane of the pixel peaks and the plane of the substrate, i.e., height differential between pin-coated and uncoated substrate areas [5].

## 4. Conclusions

In this research we show that, in contrast to highly refined fully dispersed mineral containing coating applications involving filament extension in respect to film-split, such as roll coating and printing, the increasing growth of functional coatings demands, newly, that minerals be used in their undispersed state, due to requirements of functional access to their surface properties. The study of the processability of such undispersed suspensions of minerals (without the use of dispersing stabilizers) has so far not drawn recent attention. The opposite is true for high solids content mineral slurries, for which the dispersing systems have been fully optimized. The relevance of this dearth of literature regarding undispersed systems, as the paper stresses, is no more strongly illustrated than in the recent work of Alves et al., 2020 [46], considering a formulation based on sepiolite in combination with nanocellulose fibrils, but focused exclusively on improving colloidal stability by employing a mechanical disperser and chemical dispersant, i.e., the very chemicals that we purposefully avoided in order to maintain surface activity whilst processing in a coating or printing application. One example of a hybrid printing technique becoming associated with functional coating color applications is pin coating, the suspension formulation for which has been illustrated in this paper. In pin coating, a 2D array of needle-like elements is loaded by submersion in coating suspension followed by transfer to the coating substrate by vertical contact between the pin array and the substrate surface. Loading of coating color to the pins is driven by surface tension adhesion to the surface of the pins, such that viscoelastic and shear thinning properties of suspension are, therefore, critical. The separation from the pins is driven first by surface tension adhesion of the suspension to the wettable substrate surface, followed by filament extension and breakage as the pin array is withdrawn. Thus, static yield stress needs to be overcome, firstly, so that the coating color can be transferred to the pins, and secondly, extensional viscosity controls the coating color meniscus split upon removal of the pin from the substrate surface. The findings illustrate that surface water immobilization by ancillary mineral plays a key role in developing a negatively acting highly non-linear viscoelastic rheological behavior and high extensional viscosity of coatings containing undispersed minerals, which display high yield stress and potential for dilatancy and agglomeration. These rheological limitations to process coating have been overcome with addition of micro-nanofibrillated cellulose (MNFC), which has a positive dispersive effect on the mineral particles, together with gel-like water retention and elongational orientational properties, thus reducing flocculation and dilatancy by improved lubricity during the different shear and extensional regimes that occur at the various stages of the application process. In the case exemplified, the improved processability enabled the production of a uniform pixelated coating to serve as a surface flow filter for the sorption of gaseous pollutants [4].

## Figures and Tables

**Figure 1 materials-16-01598-f001:**
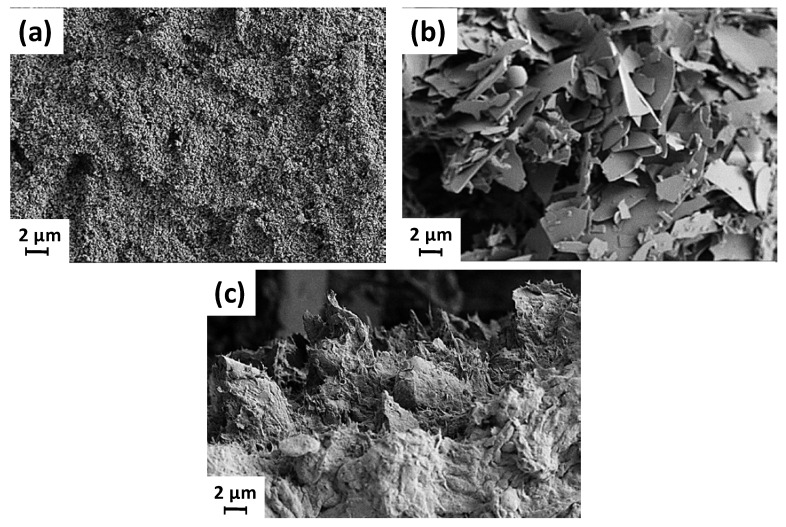
SEM image of dispersant-free particles taken from an aqueous suspension: note the flocculated structure made at 100 pph of each mineral, (**a**) GCC, (**b**) perlite, and (**c**) sepiolite.

**Figure 2 materials-16-01598-f002:**
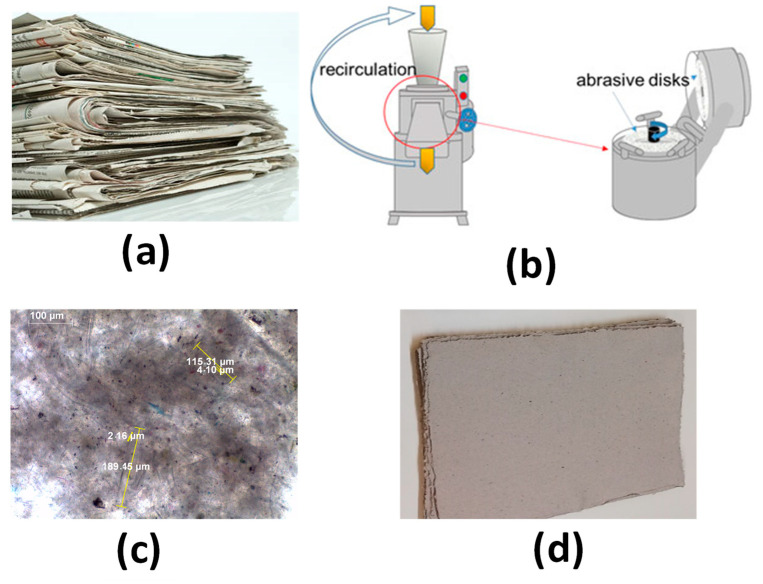
The fiber that was used from waste material, (**a**) collected newsprint was soaked in water, (**b**) schematic drawing of ultrafine super mass colloider with abrasive disks used for production of MNFC, (**c**) optical microscope image of fibrillar MNFC material obtained after grinding, showing typical MNFC fibril length and width, respectively, and (**d**) handsheets formed from a laboratory sheet former and wet pressed as substrate for filter coating [4].

**Figure 3 materials-16-01598-f003:**
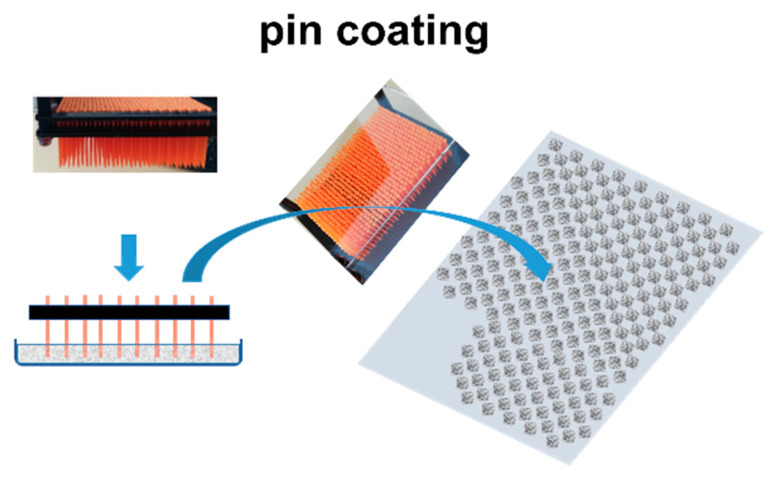
Schematic of pin coating onto a fibrous substrate (Gane et al., 2020) [4].

**Figure 4 materials-16-01598-f004:**
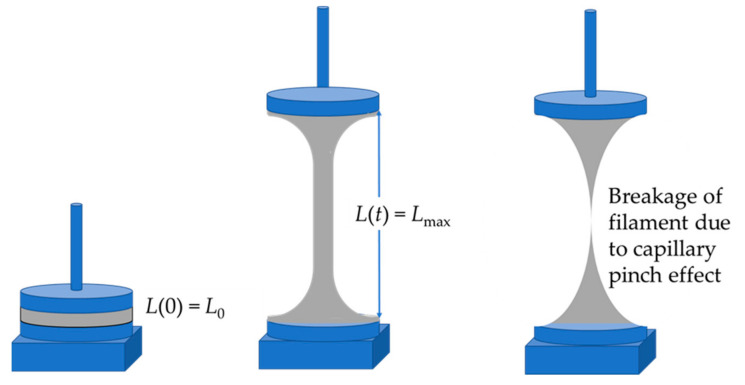
Schematic diagram of the forming filament induced by the step-stretch between the two end-plates in a CaBER experiment.

**Figure 5 materials-16-01598-f005:**
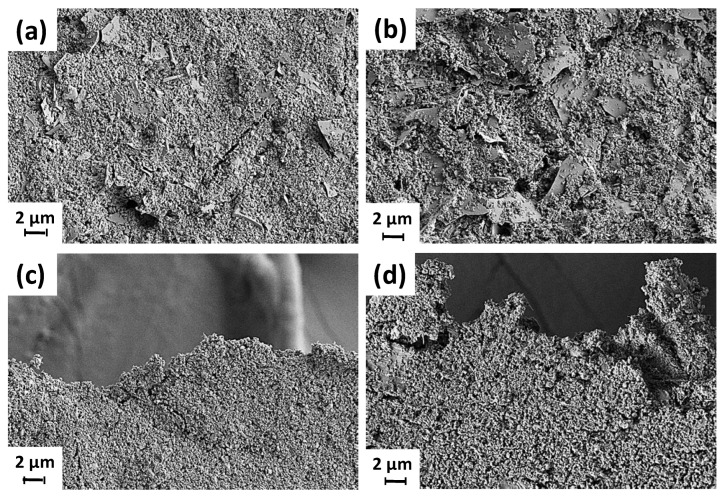
Morphology of GCC together with ancillary minerals in suspensions used in this study with partial replacement of GCC with (**a**) 5 pph perlite, (**b**) 10 pph perlite, (**c**) 5 pph sepiolite, and (**d**) 10 pph sepiolite.

**Figure 6 materials-16-01598-f006:**
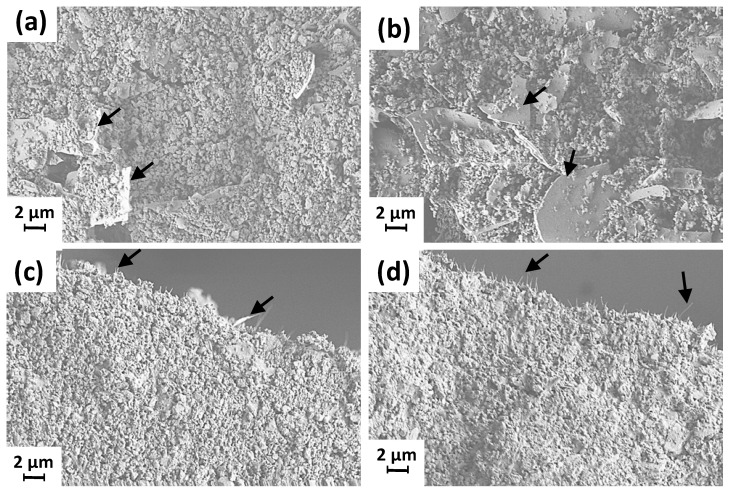
SEM images of coating colours that contained MNFC as a flow modifier, (**a**,**b**) GCC:perlite in ratio 95:5 and 90:10 (arrows indicate the fused perlite structure among the GCC particles), and (**c**,**d**) GCC:sepiolite in ratio 95:5 and 90:10 (arrows indicate the needle-like structure of sepiolite among the GCC particles).

**Figure 7 materials-16-01598-f007:**
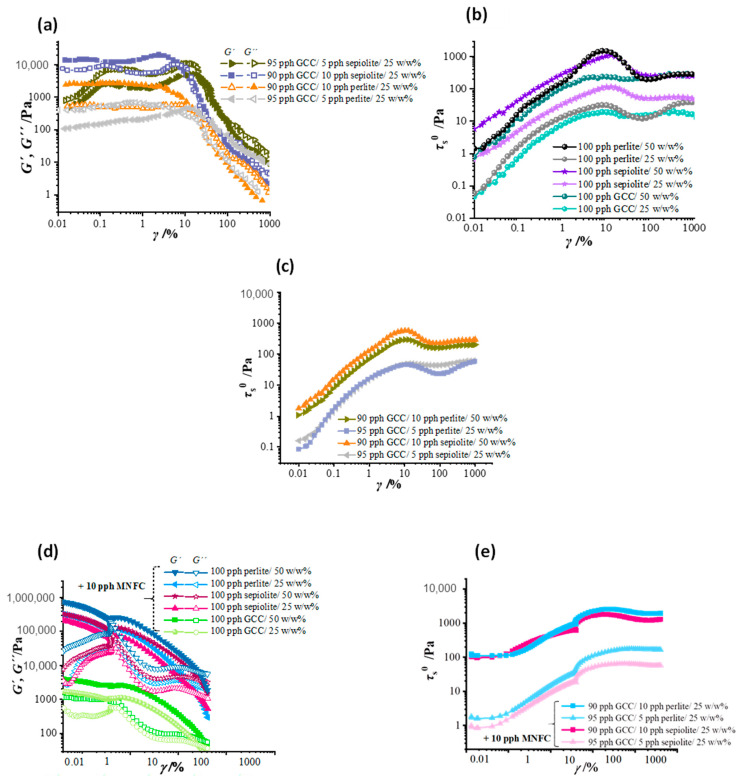
Viscoelastic response presented as (**a**) viscoelastic moduli *G*′ (solid symbols) and *G″* (open symbols) obtained from amplitude sweep tests within the quasi linear viscoelastic region (LVE) and (**b**) static stress, both as a function of strain span *γ* = 0.01 to 1 000%, revealing the different structure in respect to mineral flocculation in water suspensions, and viscoelastic initial response to increase in angular frequency *ω* = 0.01–100 (rad) s^−1^ for single pigment suspensions containing 100 pph of GCC, perlite or sepiolite, and (**c**) partial substitution of GCC by 5 pph and 10 pph of perlite and sepiolite. Strain (*γ*) sweep measurement results of coatings containing MNFC, showing (**d**) dependence of elastic moduli *G*′ and *G″* in the highly gel-like behavior of MNFC together with 100 pph of each single mineral at the two solids content of 25 *w*/*w*% and 50 *w*/*w*%, (**e**) dependence of static stress as a function of mineral particle mixture ratio together with MNFC at the lower solids content of 25 *w*/*w*%. Closed symbols denote *G*′ and open symbols *G″*.

**Figure 8 materials-16-01598-f008:**
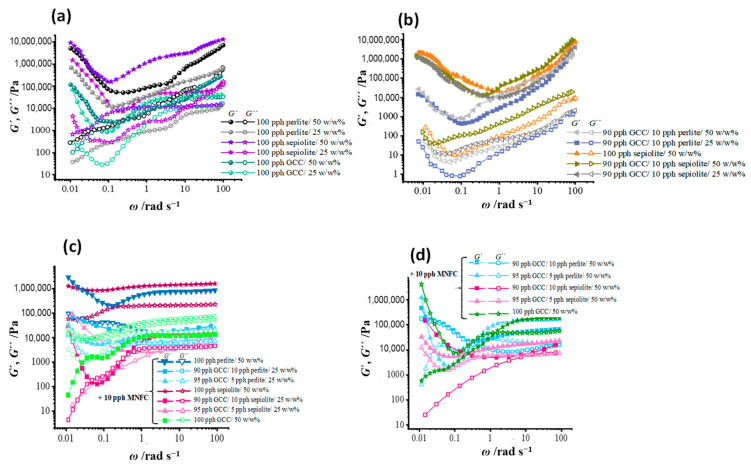
Angular frequency (*ω*) sweep response of (*G*′, *G*″) viscoelastic moduli (**a**) for the single minerals at 25 *w*/*w*% solids content, (**b**) the response to frequency for the blends at 25 *w*/*w*% and 50 *w*/*w*% solids content, and then with addition of MNFC (**c**) the response of moduli for single pigment coatings, and (**d**) the response with partial replacement of GCC with perlite and sepiolite. Closed symbols denote storage modulus (*G*′), and open symbols loss modulus (*G*″).

**Figure 9 materials-16-01598-f009:**
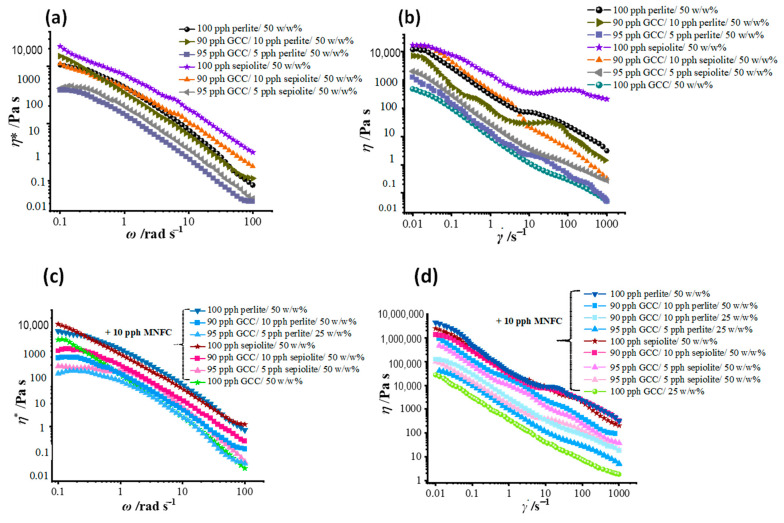
(**a**) Viscoelastic regions shown as the complex viscosity (LVE *η**) response to increase angular frequency *ω* = 0.1–100 (rad) s^−1^, and (**b**) dynamic viscosity (*η*) thinning response to shear over the range $\dot{\gamma }$ = 0.01–1000 s^−1^, the latter showing dilatant behavior at the higher shear rates. Flow modifying effect of MNFC fibrils in the (LVE) region seen as (**c**) *η** response to increase in angular frequency *ω* = 0.1–100 (rad) s^−1^, and (**d**) dynamic viscosity thinning response (*η*) over the shear rate range $\dot{\gamma }$ = 0.01–1000 s^−1^, now clearly no longer dilatant.

**Figure 10 materials-16-01598-f010:**
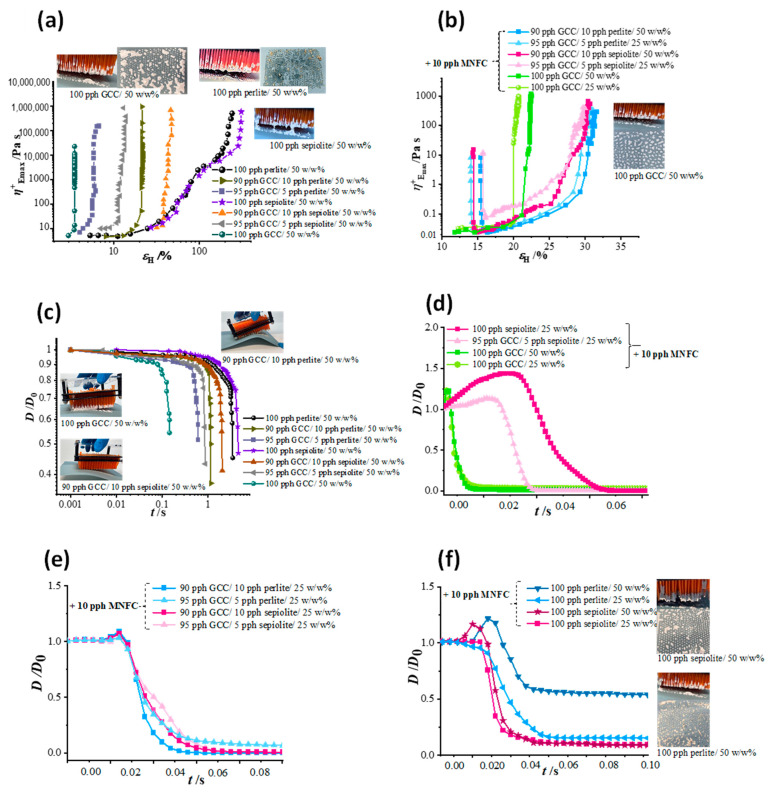
Extension of coating suspensions measured with CaBER for the GCC alone and with partial GCC replacement with ancillary mineral, up to final breaking of the filament presented as (**a**) Hencky strain (*ε*_H_) change over plate separation time, (**b**) transient extensional viscosity (*η*_E_^+^) dependence on Hencky strain (*ε*_H_) for MNFC containing coatings, and (**c**) normalized filament diameter (*D*(*t*)/*D*_0_) at the necking or pinch point for mineral-water only suspensions. Measurements of reduced filament diameter (*D*(*t*)/*D*_0_) at the necking or pinch point as a function of time in the CaBER experiments for coating in which MNFC is used as a flow modifier: (**d**) perlite and sepiolite alone, (**e**) blends of GCC with ancillary minerals at 25 *w*/*w*% solids content, and (**f**) comparison between GCC alone and coatings containing ancillary minerals.

**Figure 11 materials-16-01598-f011:**
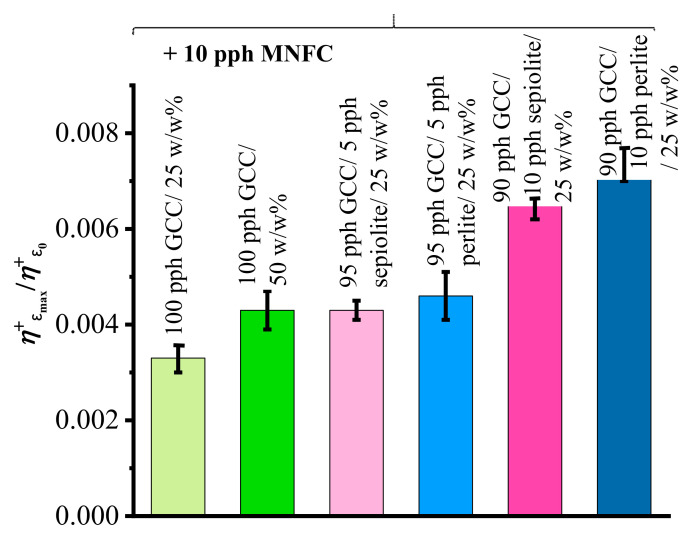
The values of transient extensional viscosity (*η^+^*_Emax_) at maximum Hencky strain (*ε*_Hmax_) when filament break-up occurs, reduced with initial values of (*η^+^*_E0_) at the start of the plate separation regime during CaBER measurements for MNFC containing coating suspensions.

**Figure 12 materials-16-01598-f012:**
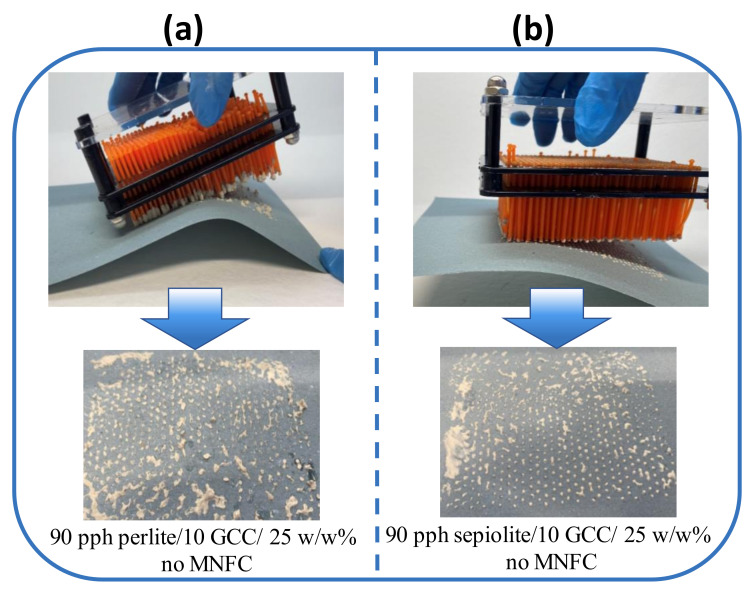
Images of pin coating process where coating color medium is sandwiched between application pins and substrate of recycled newsprint, including the complex interaction of viscoelastic bahavior, shear response and extensional flow, affecting transfer of the positioned pixel array. Coating color transfer of 10 pph ancillary minerals in blend with 90 pph GCC matrix creating uneven dot shape and poor separation from pin to substrate with (**a**) sepiolite and (**b**) perlite as ancillary minerals in water only suspension, and with the contrasting micro nanofibrillar cellulose (MNFC) dispersion effect for coating color (**c**) containing sepiolite and (**d**) containing perlite.

**Table 1 materials-16-01598-t001:** Particle size upper limits, *d* (vol), in the fractions of GCC contained in the <10 *v*/*v*%, <50 *v*/*v*% (median), and <90 *v*/*v*% regions before and after ultrasonication in water [4].

Volume Defined Particle Size Distribution Measured by Light Scattering		
Particle Size	Before Sonication/μm	After Sonication/μm
*d*_10_ (vol)	3.33	1.12
*d*_50_ (vol)	4.61	1.41
*d*_90_ (vol)	6.05	1.77

**Table 2 materials-16-01598-t002:** Experimental mineral water suspensions and coating colors. All formulations were additionally studied with 10 pph MNFC as a modifier *.

Mineral Component/pph * (+ 10 pph MNFC)	Solid Content/*w*/*w*%	Formulation Nomenclature
100 GCC	25	100 pph GCC/25 *w*/*w*%
	50	100 pph GCC/50 *w*/*w*%
100 perlite	25	100 pph perlite/25 *w*/*w*%
	50	100 pph perlite/50 *w*/*w*%%
100 sepiolite	25	100 pph sepiolite/25 *w*/*w*%
	50	100 pph sepiolite/50 *w*/*w*%
95:5 GCC:perlite	25	95 pph GCC/5 pph perlite/25 *w*/*w*%
	50	95 pph GCC/5 pph perlite/50 *w*/*w*%
95:5 GCC:sepiolite	25	95 pph GCC/5 pph sepiolite/25 *w*/*w*%
	50	95 pph GCC/5 pph sepiolite/50 *w*/*w*%
90:10 GCC:perlite	25	90 pph GCC/10 pph perlite/25 *w*/*w*%
	50	90 pph GCC/10 pph perlite/50 *w*/*w*%
90:10 GCC:sepiolite	25	90 pph GCC/10 pph sepiolite/25 *w*/*w*%
	50	90 pph GCC/10 pph sepiolite/50 *w*/*w*%

**Table 3 materials-16-01598-t003:** Rheological parameters of coating colors formulated for pin coating from GCC, perlite and sepiolite, and mixes thereof, comparing the cases with and without flow modifier MNFC made from waste newsprint.

Sample	Flow Modifier MNFC /pph	*n* _c_	*n*	*k*_c_ /Pa s*^n^*^c^	*k* /Pa s*^n^*	*τ*_s_^0^ /Pa	*τ*_d_^0^ /Pa
100 pph GCC/25 *w*/*w*%	10	0.89	0.86	21.4	19.5	164	148
95 pph GCC/5 pph perlite/25 *w*/*w*%	10	0.89	0.84	51.4	46.8	255	251
95 pph GCC/5 pph sepiolite/25 *w*/*w*%	10	0.87	0.84	96.1	87.4	270	257
90 pph GCC/10 pph perlite/25 *w*/*w*%	10	0.77	0.75	101.6	92.4	312	285
90 pph GCC/10 pph sepiolite/25 *w*/*w*%	10	0.70	0.68	106.9	97.2	334	321
100 pph perlite/25 *w*/*w*%	10	0.66	0.65	127.7	116.1	423	387
100 pph sepiolite/25 *w*/*w*%	10	0.64	0.63	148.1	134.6	467	432
100 pph GCC/50 *w*/*w*%	10	0.64	0.62	160.6	146.2	537	485
95 pph GCC/5 pph perlite/50 *w*/*w*%	10	0.63	0.61	201.7	182.8	579	527
95 pph GCC/5 pph sepiolite/50 *w*/*w*%	10	0.79	0.79	219.1	199.1	601	548
90 pph GCC/10 pph perlite/50 *w*/*w*%	10	0.76	0.76	269.1	244.6	622	569
90 pph GCC/10 pph sepiolite/50 *w*/*w*%	10	0.72	0.72	295.5	268.6	687	638
100 pph perlite/50 *w*/*w*%	10	0.71	0.70	307.2	279.3	1334	1248
100 pph sepiolite/50 *w*/*w*%	10	0.70	0.69	322.3	293.2	2234	2178
100 pph GCC/25 *w*/*w*%	0	0.69	0.67	43.4	39.5	867	739
95 pph GCC/5 pph perlite/25 *w*/*w*%	0	0.66	0.66	117.4	106.8	1210	943
95 pph GCC/5 pph sepiolite/25 *w*/*w*%	0	0.65	0.63	162.1	147.4	1264	1195
90 pph GCC/10 pph perlite/25 *w*/*w*%	0	0.61	0.59	189.6	172.4	1365	1375
90 pph GCC/10 pph sepiolite/25 *w*/*w*%	0	0.58	0.56	216.9	197.2	1421	1494
100 pph perlite/25 *w*/*w*%	0	0.67	0.66	259.7	236.1	1576	1468
100 pph sepiolite/25 *w*/*w*%	0	0.66	0.62	291.1	264.6	1584	1543
100 pph GCC/50 *w*/*w*%	0	0.64	0.62	325.7	296.1	2290	1724
95 pph GCC/5 pph perlite/50 *w*/*w*%	0	0.61	0.59	353.4	321.3	2344	1892
95 pph GCC/5 pph sepiolite/50 *w*/*w*%	0	0.73	0.72	426.1	387.4	2568	2034
90 pph GCC/10 pph perlite/50 *w*/*w*%	0	0.67	0.66	535.9	487.2	2816	2132
90 pph GCC/10 pph sepiolite/50 *w*/*w*%	0	0.68	0.64	588.1	534.6	3131	2633
100 pph perlite/50 *w*/*w*%	0	0.64	0.62	845.1	768.2	3598	2882
100 pph sepiolite/50 *w*/*w*%	0	0.57	0.55	1092.3	993.4	4175	3552

## Data Availability

Data are available on demand from corresponding author.

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
