# Peer review of "Micro Nanofibrillated Cellulose as Functional Additive Supporting Processability of Surface-Active Mineral Suspensions: Exemplified by Pixel Coating of an NOx-Sorbent Layer"

_materials, 2023, doi:10.3390/ma16041598_

Round 1

Reviewer 1 Report

The manuscript entitled “Micro nanofibrillated cellulose as functional additive supporting processability of surface-active mineral suspensions: exemplified by pixel coating of an NOx-sorbent layer” requires minor revision before publication. The author mainly described the rheological behavior of MNFC for pin coating. The works in the mechanical analysis and coating performance are sound to clarify practical conditions. Although the manuscript is ready to be accepted, several suggestions were made to authors for revision. Please find the comments below.

 (1) The English usage in the manuscript requires improvements or suggestions from native speakers. Many sentences are confused, for example, at the line 139-140.

In contrast, sepiolite effects COcapture via an adsorption mechanism relying on the function of its surface –OH groups leading to acid-base intercalation of CO2, Eq. (2)

(2) The definition of calcium carbonate (GCC) at line 111 is confused to the GCC at line 16.

(3) Between the lines 131 and 136, the format doesn’t different to the context.

(4) What is the Thomson (Lord Kelvin) effect? Is it relevant to the research?

(5) Please explain the dimension marked in Figure 2c? Fibrillar MNFC’s length and width?

(6) What is the reference number of Figure 3?

(7) The notation of Figure 5c and 5d is the same. Is it wrong?

(8) Equation 7~11 isn’t well discussed in the “Results and discussion” although they are well listed in the section of “Materials and methods”.

Author Response

Responses to Reviewer Report 1

Thanks are given to the reviewer for commenting positively on our contribution in this area of research.

(1)         The English usage in the manuscript requires improvements or suggestions from native speakers. Many sentences are confused, for example, at the line 139-140.

In contrast, sepiolite effects CO2 capture via an adsorption mechanism relying on the function of its surface –OH groups leading to acid-base intercalation of CO2, Eq. (2)

Firstly, we have undertaken that the manuscript has been fully reviewed and edited by a native English speaking scientist.

A suggestion to make the quoted sentence clearer is as follows:

In contrast, sepiolite effects the capture of CO2 by adsorption via the function of surface –OH groups leading to acid-base intercalation, Eq. (2).

(2)         The definition of calcium carbonate (GCC) at line 111 is confused to the GCC at line 16

The confusion has been remedied by inserting the term “ground” once again at the point of the second description.

(3)         Between the lines 131 and 136, the format doesn’t differ from the context.

This was an unfortunate effect due to the insertion of the Table above. Now corrected, thank you.

(4)         What is the Thomson (Lord Kelvin) effect? Is it relevant to the research?

To be more precise we refer to the Joule-Thomson effect, which is a condensation of gas to liquid when confined in a microscopic volume, such as a capillary. It is relevant as a method for capturing gas whilst flowing through a porous medium, where, on entering a fine pore, it can undergo condensation. This phenomenon is now explained in the text as follows:

Joule-Thomson (Lord Kelvin) effect, whereby a confined gas condenses to a liquid at a chemical potential below that corresponding to liquid–vapor coexistence in the bulk.

(5)         Please explain the dimension marked in Figure 2c. Fibrillar MNFC’s length and width?

The assumption by the reviewer is correct. The caption to Figure 2 has been modified to read:

….showing typical MNFC fibril length and width, respectively, ……..

(6)         What is the reference number of Figure 3?

Thank you for spotting this omission. It is reference 4, and this has been added into the caption.

(7)         The notation of Figure 5c and 5d is the same. Is it wrong?

Thank you. Naturally it should have been “10 pph”. This has now been corrected.

(8)         Equations 7-11 are not well discussed in the “Results and discussion” although they are well listed in the section of “Materials and methods”.

The equations 10-11 were included simply to complete the theory for multimodal behavior. We agree that these are not necessary to follow the observations in the Results and discussion section, and so they have been removed. The observations rely only on the ratio D(t)/D0, depending on Equations 7-9 alone.

In combination with this redundancy, Regime II has been removed from Figure

Thank you once again for helpful commentary on the manuscript.

Reviewer 2 Report

The manuscript can be accepted for publication after addressing the following issues.

1. I suggest that the author add some relevant references from recent years.

2. The author should explain why 10 pph MNFC is used and not other levels. The loading of the MNFC has a large influence on the properties of the suspension, especially the rheological properties.

3. Did the authors characterize the microscopic topography of MNFC and obtain some parameters? Such as diameter, length-diameter ratio, etc. These parameters have a great influence on the rheological properties.

Author Response

Responses to Reviewer Report 2

Thanks to the reviewer for the insightful suggestions.

Suggestions for authors

  1. I suggest that the author add some relevant references from recent years.

Thank you for the suggestion: some of the authors are experts in mineral suspensions and coatings using them, and, therefore, have ready access to a very wide range of classic and recent studies, however, as the paper stresses, the study of undispersed suspensions of minerals (no use of dispersing stabilizers) has not had any recent attention. The opposite is true for high solids content mineral slurries for which the dispersing systems have been fully optimized. The move toward pure functionality, in respect to surface activation of minerals in a coating context, is currently a neglected area of processability study, and so there is a dearth of recent literature. We have confirmed this by making a second round of literature search, and have generally drawn a blank. One good find, though, which has now been included is work by Alves et al. Improving Colloidal Stability of Sepiolite Suspensions: Effect of the Mechanical Disperser and Chemical Dispersant. Minerals 2020, 10(9), 779; https://doi.org/10.3390/min10090779, in which an example is given of improving sepiolite dispersion in combination with cellulose nanofibrils by addition of chemical dispersant, the very chemicals in fact that we tried to avoid to maintain surface activity, i.e. we can use this as a good counter argument case to show how the processing without dispersant is indeed a challenge, and one that the work reported in the manuscript undertook. This has now been included as a supporting reference in the Conclusions section by the following:

The study of the processability of such undispersed suspensions of minerals (without the use of dispersing stabilizers) has so far not drawn recent attention. The opposite is true for high solids content mineral slurries, for which the dispersing systems have been fully optimized. The relevance of this dearth of literature regarding undispersed systems, as the paper stresses, is no more strongly illustrated than in the recent work of Alves et al., 2020, 44 considering a formulation based on sepiolite in combination with nanocellulose fibrils, but focused exclusively on improving colloidal stability by employing a mechanical disperser and chemical dispersant, i.e., the very chemicals that we purposefully avoided in order to maintain surface activity whilst processing in a coating or printing application.

  1. The author should explain why 10 pph MNFC is used and not other levels. The loading of the MNFC has a large influence on the properties of the suspension, especially the rheological properties.

We fully agree with this limitation in respect to exploring a range of loading levels of MNFC. It is to be fully expected that the nature of a specific application will demand an optimal use of the flow modifier chosen, such as MNFC. The focus of the work described in the paper, however, was to illustrate the development of a formulation for the example application of pin coating a substrate to form a surface-flow gas filter. The application was specific, in that the capture of the NOx target gas on calcium carbonate whilst arresting the release of the in-situ formed CO2 demanded a defined simultaneous action of binding and humectant effect derived from the MNFC. Less than 10 pph MNFC simply resulted in insufficient binding to the substrate. This is now explained in the text.

It is to be expected that the nature of a specific application will demand an optimal use of the additive flow modifier chosen, such as MNFC. The focus of the work described in the paper is to illustrate the development of a formulation containing undispersed mineral mixtures for the example application of pin coating onto a substrate to form a surface-flow gas filter. The application is, therefore, specific, in that the capture of the NOx target gas on calcium carbonate, whilst simultaneously arresting the release of the in-situ formed CO2, demanded a defined triple action of water retention, binding and humectant effect derived from the MNFC. Though limiting any exploration of the rheological effect over a range of addition levels for MNFC, it was found in practice that less than 10 pph MNFC simply resulted in insufficient binding to the highly porous substrate, as well as being insufficient to retain the necessary moisture to drive the desired reaction in Eq. (1) under possibly low humidity conditions. For these reasons the illustrated study is confined to the chosen level of 10 pph MNFC.

  1. Did the authors undertake to characterize the microscopic topology of MNFC and obtain some parameters? Such as diameter, length-diameter ratio etc. These parameters have a great influence on the rheological properties.

The morphology of the MNFC fibrils is a very important factor, we agree. We support this by providing the microscopic analysis illustrated in Figure 2(c), in which the typical length and width of the fibrils are shown, and supported now in the text as follows:.

Typical MNFC fibril morphology, together with example length and width dimensions, are visible in detail in Figure 2(c).